# The Neutrino Mass Problem: From Double Beta Decay to Cosmology †

**Osvaldo Civitarese**

Department of Physics, Institute of Physics La Plata, University of La Plata, La Plata 1900, Argentina; osvaldo.civitarese@fisica.unlp.edu.ar
† Dedicated to Sabin Stoica 70 anniversary.

**Abstract:** The neutrino is perhaps the most elusive member of the particle zoo. The questions about its nature, namely: Dirac or Majorana, the value of its mass and the interactions with other particles, the number of its components including sterile species, are long standing ones and still remain to a large extent without conclusive answers. From the side of the nuclear structure and nuclear reactions, both theories and experiments, the need to elucidate these questions has, and still has, prompt crucial developments in the fields of double beta decay, double charge exchange and neutrino induced reactions. The measurements of neutrino flavor oscillation parameters contribute largely to restrict models with massless neutrinos. From the particle physics side, the possibilities to extend the standard model of electroweak interactions to incorporate a right-handed sector of the electroweak Lagrangian are directly linked to the adopted neutrino model. Here, I would like to address another aspect of the problem by asking the question of the neutrino mass mechanism in the cosmological context, and particularly about dark matter.

**Keywords:** neutrino mass; axions; neutrino-axion couplings; double beta decay





## 1. Introduction

The theoretical description of double beta decay modes requires the use of techniques that originate in various branches of physics, such as: (a) nuclear interactions, (b) nuclear structure models and methods, (c) electroweak decays, and (d) elementary particle physics models and related symmetries.

At first glance of a different field, i.e., the composition of the dark matter in the universe, questions arose that are closely related to the double beta decay studies. Among these questions is the one related to the origin of the neutrino mass.

This paper is devoted to the exploration of the possibilities of the assumption about the existence of a neutral-bosonic-complex scalar field, called the axion, offer to the theory of electroweak interactions.

Some of these questions have been addressed long ago by John D. Vergados in several pioneering papers [1–5]. Vergados's papers described in detail the role of axions in the composition of dark matter. He also provided very detailed results about detection signals due to modulation effects caused by the presence of dark matter.

In this paper, we shall focus on the particular case of the neutrino mass by studying the coupling of neutrinos with axions.

The conventional Higgs mechanism does not give mass to neutrinos, as it does with other particles. Some 50 years ago, R.D. Peccei and H.R. Quinn [6,7] have proposed the existence of the axion in order to explain for the suppression of the neutron electric-dipole moment, and introduced a global U(1) symmetry called the Peccei–Quinn (PQ) symmetry. In this picture, the physical vacuum contains some background fields $\Phi$, being the axion one of these. In the simplest version, the axion is a neutral scalar complex field that acquires a non-vanishing vacuum expectation value due to its associated double well potential [6,7].

Shortly after the Big Bang, the temperature was high and the PQ symmetry was manifest. When the temperature fell low enough, a phase transition occurred and the PQ field acquired a non-zero vacuum expectation value $\langle\Phi\rangle_0$, leading to a spontaneous symmetry breaking at an unknown energy scale $f_a$ [7].

The coupling of the axion to the gluons and to quarks suppresses the neutron electric-dipole moment, providing a solution to the strong CP problem. In extended scenarios of the couplings, the axion could also interact with pairs of photons and pairs of baryons.

In addition to their role in cosmology, with reference to the dark matter composition [4,5,8], axions may play a role in neutrino physics [9], because the coupling of neutrinos with axions could provide a mechanism to explain non-zero neutrino masses.

In analogy to the conventional Higgs's mechanism, the addition of a neutral scalar field with a non-zero vacuum expectation value in a quadratic plus quartic potential provides a mass term when coupled to neutrinos, in the same way the non-zero vacuum expectation value of the Higgs's boson gives mass to other particles. Naturally, the axion–neutrino coupling is not responsible for the neutrinoless double beta decay mode, rather it gives mass to the neutrino.

The dynamics of the double beta decay includes other no-less important ingredients, as well, both from the nuclear structure side, such as availability in phase space, the inclusion of nucleon and nuclear correlations and the knowledge of the microscopic structure of participant nuclear states and their energy differences.

Additionally, it requires the extension of the Standard Model Lagrangian by the inclusion of right-handed currents and their couplings [10].

With this motivation in mind, we have explored the consequences of the coupling between axions and neutrinos, in order to compare the neutrino mass values resulting from this coupling with the upper limits to the neutrino mass determined from the non-observation of the neutrinoless double beta decay. We have introduced a Lagrangian describing the neutrino–axion coupling and calculated the neutrino mass insertion. Next, we have reviewed briefly the essentials of the formalism of neutrinoless double beta ($0\nu\beta\beta$) decay to make a connection with the axion–neutrino picture and compared the results from both scenarios to investigate the compatibility between them.

Some of the results that we are going to present here can be found in the works of Refs. [11–15], where the details of the calculations have been presented.

## 2. About the Formalism

We start from the Lagrangian [16,17]

$$\mathcal{L}_{int} = ig_{av}\bar{\psi}\gamma^{\mu}\gamma^{5}\psi\partial_{\mu}\Phi \tag{1}$$

which describes the derivative coupling between neutrinos ($\psi$) and axions ($\Phi$).

By separating spatial and temporal derivatives, the Lagrangian is split up in the following terms:

$$\mathcal{L}_{int} = ig_{av}\psi^{\dagger}\vec{\sigma}\psi \cdot \vec{\nabla}\Phi + ig_{av}\psi^{\dagger}\gamma^{5}\psi\partial_{0}\Phi. \tag{2}$$

Following the argument by Peccei and Quinn [6,7], the axion field acquires a non-zero vacuum expectation value, $\langle\Phi\rangle_0$, in presence of the potential [16,17] [1]

$$V(\Phi) = -\frac{\mu^2}{2}\left(|\Phi|^2 - \frac{1}{f_a^2}|\Phi|^4\right). \tag{3}$$

By imposing the condition

$$\left.\frac{\partial V}{\partial \Phi}\right|_{\Phi=\langle\Phi\rangle_0} = 0, \tag{4}$$

one obtains the solutions

$$\langle\Phi\rangle_0 = 0 \quad (\text{unstable point}), \tag{5}$$

and

$$|\langle\Phi\rangle_0| = \frac{f_a}{\sqrt{2}}. \tag{6}$$

Thus, the axion scalar field is written

$$\Phi \to \Phi(\vec{x}, t) + \langle\Phi\rangle_0, \tag{7}$$

and with it we obtain for the Lagrangian, written in natural units, the expression:

$$\mathcal{L}_{int} \approx g_a |\langle\Phi\rangle_0| \psi^\dagger\psi + g_{av}\Phi(\psi^\dagger\vec{\sigma}\psi)\cdot\vec{p} \tag{8}$$

Therefore, at the lowest order in the neutrino–axion interaction, we introduce the correspondence

$$m_\nu \to g_a|\langle\Phi\rangle_0| \tag{9}$$

since $\psi^\dagger\psi$ is the neutrino density.

To calculate the contributions to the neutrino mass coming from the spin-dependent term of the Lagrangian, we write, for the transition amplitude [11]

$$\mathcal{A}_{i\to f} = \langle f|\mathrm{T}\left\{(-i)\int d^4x\hat{\mathcal{H}}_{int}(x)\right\}|i\rangle = -ig_{av}\int d^4x \langle f|\vec{\nabla}\Phi\cdot\vec{\mathbf{S}}|i\rangle, \tag{10}$$

where $\vec{\mathbf{S}}$ is acting on the fermionic sector.

For spin-up and for spin-down neutrino states, we obtain:

$$
\begin{aligned}
\langle f|\vec{\nabla}\Phi\cdot\vec{\mathbf{S}}|i\rangle &= i\mathcal{N}_i\mathcal{N}_f\left[\left(1 + \frac{(p'_z p_z - p'_- p_+)}{(E+m)(E'+m)}\right)\frac{\partial\Phi}{\partial z}\right.\\
&\quad + \frac{(p'_- p_z + p'_z p_+)}{(E+m)(E'+m)}\frac{\partial\Phi}{\partial x}\\
&\quad + \left. i\frac{(-p'_z p_+ + p'_- p_z)}{(E+m)(E'+m)}\frac{\partial\Phi}{\partial y}\right]
\end{aligned}
\tag{11}
$$

and

$$
\begin{aligned}
\langle f|\vec{\nabla}\Phi\cdot\vec{\mathbf{S}}|i\rangle &= -i\mathcal{N}_i\mathcal{N}_f\left[\left(1 + \frac{(p'_z p_z - p'_+ p_-)}{(E+m)(E'+m)}\right)\frac{\partial\Phi}{\partial z}\right.\\
&\quad + \frac{(p'_+ p_z + p'_z p_-)}{(E+m)(E'+m)}\frac{\partial\Phi}{\partial x}\\
&\quad + \left. i\frac{(p'_+ p_z - p'_z p_-)}{(E+m)(E'+m)}\frac{\partial\Phi}{\partial y}\right],
\end{aligned}
\tag{12}
$$

respectively, where for short $\Phi = \Phi(\vec{x}, t)$ in the above equations. These expressions are model-dependent since they would imply the knowledge of the spatial dependence of $\Phi(\vec{x}, t)$.

For the sake of completeness, as an example of spatial DM distributions, if we adopt for the axions a directional Gaussian parallel to the neutrino incoming direction (arbitrarily chosen in the z-direction), there will not be a spin-flip term contributing to the amplitude $\langle f|\vec{\nabla}\Phi.\vec{\mathbf{S}}|i\rangle$, while the spin-up contribution will then look like [11–13]

$$\langle f|\vec{\nabla}\Phi\cdot\vec{\mathbf{S}}|i\rangle = i\mathcal{N}_i\mathcal{N}_f\left(1 + \frac{(p'_z p_z - p'_+ p_-)}{(E+m)(E'+m)}\right)\frac{\partial\Phi}{\partial z}. \tag{13}$$

and for the spin-down one we have

$$\langle f|\vec{\nabla}\Phi \cdot \vec{\mathbf{S}}|i\rangle \;\;=\;\; -i\mathcal{N}_i\mathcal{N}_f\left(1+\frac{(p'_z p_z - p'_+ p_-)}{(E+m)(E'+m)}\right)\frac{\partial \Phi}{\partial z}. \tag{14}$$

In Equations (13) and (14), for the matrix elements $\langle f|\vec{\nabla}\Phi \cdot \vec{\mathbf{S}}|i\rangle$, $\mathcal{N}_i$ and $\mathcal{N}_f$ are the normalization factors of the initial and final neutrino states, respectively. The quantities with primes define the energy and components of the neutrino momentum in the final state.

The corrections $(S'(p))$ to the bare fermion (neutrino) propagator $(S(p) = (p-m)^{-1})$ [16,17], due to interactions with axions, are analytically expressed as [14]

$$S'(p) \;\;=\;\; S(p) + S(p)(\imath\Sigma(p))S(p) + \dots \tag{15}$$

After evaluating $\Sigma(p)$ on shell, we have finally obtained the one-loop correction to the effective neutrino mass due to the interaction with axions. It is written as

$$\Sigma(p) \;\;=\;\; \frac{g_a^2}{8\pi^2}\left(p\cdot\Sigma_p + m\Sigma_m\right), \tag{16}$$

The explicit forms of $\Sigma_p$ and $\Sigma_m$ are listed in [15],

The effective mass of the neutrino can be computed as

$$m_\nu \;\;=\;\; m + \Sigma(p)\Big|_{p^2=m^2} \tag{17}$$

To eliminate divergent contributions in the loop expansion, we have defined a mass $\tilde{m}_\nu$,

$$\tilde{m}_\nu \;\;=\;\; m\left[1+\frac{g_a^2}{8\pi^2}\frac{3}{\epsilon}\right], \tag{18}$$

as explained in Refs. [14,15], and expressed the effective neutrino mass, for the electron neutrino flavor, in terms of it.

The effective neutrino mass is finally written as

$$\begin{aligned}\frac{m_\nu}{\tilde{m}_\nu}-1 \;\;=\;\; & \frac{g_a^2}{8\pi^2}\Bigg\{-\frac{3}{2}\gamma+2+\frac{1}{2}\frac{m_a^2}{m_\nu^2}-\frac{3}{2}\ln\left(\frac{m_\nu^2}{4\pi\xi^2}\right)+\frac{1}{4}\frac{m_a^4}{m_\nu^4}\ln\left(\frac{m_\nu^2}{m_a^2}\right)\\ & -2\frac{m_a}{m_\nu}\sqrt{\beta}\left[\frac{m_a^2}{2m_\nu^2}\mathrm{Arctg}\left(\frac{m_a}{m_\nu}\sqrt{\beta}\right)+\mathrm{Arctg}\left(\frac{m_\nu}{m_a}\frac{\zeta}{\sqrt{\beta}}\right)\right.\\ & \left.-\frac{\zeta}{2}\mathrm{Arctg}\left(\frac{m_\nu}{m_a}\sqrt{\beta}\right)\right]\Bigg\}. \end{aligned} \tag{19}$$

The derivation of the previous equations involved the ordering of higher order corrections to the propagator, as well as the definition of a criteria to determine the strength of the coupling $g_a$ for each proposed mass scale $m_a$. The details have been presented in Refs. [11–15] [2].

*Neutrinoless Double-Beta-Decay Rates*

Limits to the neutrino mass can be set by comparing the theoretical rates and the experimental limits for the non-observation of the neutrinoless double beta decay. The half-life of the neutrinoless double beta decay is written in the left-right representation [18]

$$
\begin{aligned}
t_{1/2}^{-1} \;=\;\; & C_{mm}\left(\frac{m_\nu}{m_e}\right)^2 + C_{m\lambda}<\lambda>\left(\frac{m_\nu}{m_e}\right) \\
+\;\; & C_{m\eta}<\eta>\left(\frac{m_\nu}{m_e}\right) + C_{\lambda\lambda}(<\lambda>)^2 \\
+\;\; & C_{\eta\lambda}<\lambda><\eta> + C_{\eta\eta}(<\eta>)^2 \;,
\end{aligned}
\tag{20}
$$

where the coefficients $C_{ij}$ are functions of the nuclear matrix elements and couplings corresponding to the mass, left-handed, right-handed and cross terms of the current–current interactions appearing in models beyond the Standard Model [8,9,18]. These coefficients, ($C_{ij}$), where the sub-indexes $i(j)$ denote contributions coming from products of left- and right-handed terms of the current–current interactions, include products of the components U and V of the neutrino mixing matrices, nuclear matrix elements of Gamow–Teller and Fermi-type operators between the participant nuclear states and phase space integrals. Tables with the values of these coefficients for different double beta decay systems are listed in Ref. [18].

The effective (flavor) neutrino mass appearing in the previous equation contains the factors determined from the observation of neutrino flavor oscillations. They are functions of the amplitudes $U_{ej}$ and of the square mass differences between neutrino mass eigenstates $m_j$, as well as of phases coming from symmetry violations in the neutrino sector. In the present case, we shall take the value $m_\nu$ as the effective neutrino mass, for the electron neutrino flavor, and compare its value with the one determined from the coupling with axions given by Equation (19).

### 3. Some Results

As a first step, we have computed the maximum value for the neutrino mass considering the limits imposed by the non-observation of the neutrinoless double beta decay.

The value of $m_\nu$ extracted from the limits on the half-life (20) is model dependent, since the assumptions on nuclear structure and leptonic phase space factor are contained in the coefficients $C_{ij}$, as explained before. For the sake of the present calculations, we have taken their values from Ref. [18]. The maximum average value for the electron neutrino mass is approximately 0.3 eV and corresponds to $<\lambda>=<\eta>=0$.

Afterwards, we have calculated one-loop corrections to the neutrino-mass propagator as a function of the axion mass and of the coupling constant $g_a$.

We have performed two different analysis, namely:

- (1): by fixing the value of $\tilde{m}_\nu$ at the zero-order neutrino mass (cf. Equation (9)). In this manner, the bare value of the neutrino mass ($m$) of Equation (18) is also fixed;
- (2): by fixing the value of $\tilde{m}_\nu$ at the maximum value $m_\nu$ allowed by the non-observation of the neutrinoless double beta decay. This is done by taking the average resulting from the theoretical estimates of the ratios between the half-lives and the coefficients $C_{mm}$ for the double beta decay emitters listed in Ref. [18]. This choice of $\tilde{m}_\nu$ is arbitrary, because of the uncertainties affecting the values of the different nuclear matrix elements, but it should be taken as a demonstration of the feasibility of the method.

In both cases, we have calculated the effective neutrino mass for different values of the axion mass $m_a$ and of the coupling constant $g_a$. The results have been presented in Ref. [14]. To give an example of the method that we have developed, we show in Figure 1 the dependence of the electron neutrino mass, as a function of $m_a$ and $g_a$. In spite of the rather involved structure exhibited by the propagator's corrections, the results show a smooth trend for $m_\nu$ as a function of $m_a$.

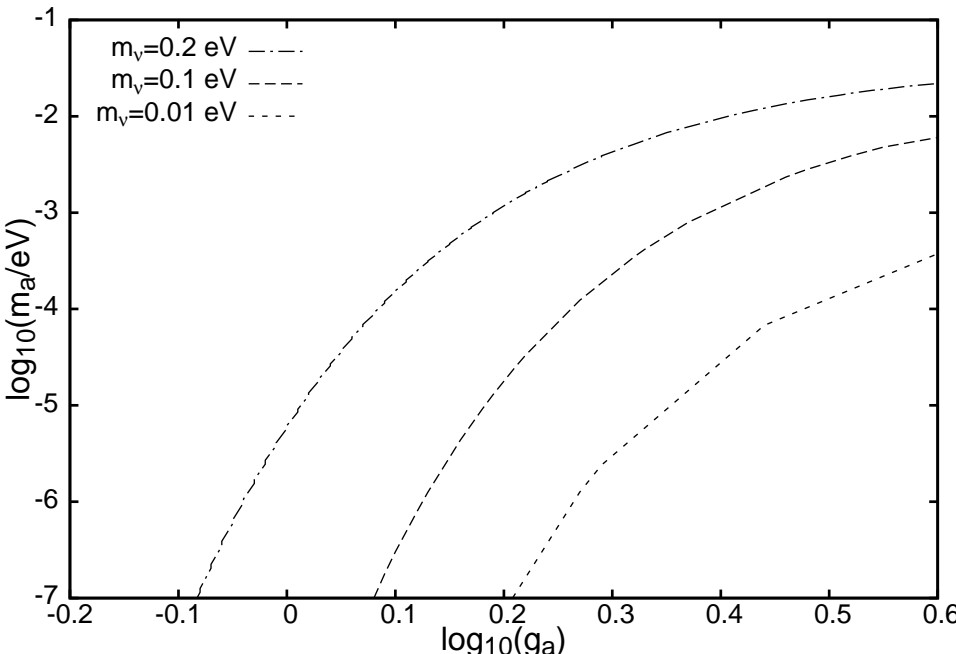

**Figure 1.** The effective neutrino mass $m_\nu$, as a function of $m_a$ and $g_a$. The results of Equation (19) are given for the largest value of $\tilde{m}_\nu$ (0.3 eV).

We can the summarize the results in the following:

- The non-observation of the neutrinoless double beta decay provides limits on the effective neutrino mass $m_\nu$, which are compatible with values determined from the coupling of neutrinos with axions.
- The values ($m_a$ and $g_a$), which are consistent with $m_\nu$, can be taken as reference values for dark-matter studies.

## 4. Conclusions

The double beta decay, in its neutrinoless mode, is perhaps one of the most rich processes from the point of view of the physics involved. Its observation will definitively demonstrate that the current view of the electroweak interactions must be changed drastically. It will also determine the future of large scale experiments, such as ATLAS in CERN, in the search of new generations of mediators of the weak interactions, both in the bosonic and fermionic sectors of the theory. From the point of view of nuclear structure models, it will also set a very selective criteria for the adoption or rejection of nuclear Hamiltonians, nuclear coupling schemes and methods to determine nuclear wave functions. Last but not least it will demonstrate the validity of Majorana's theory of the neutrino.

As an alternative way of thinking about double beta decay and neutrino properties, we have extended the notion of the U(1) symmetry breaking in the axion sector of a Lagrangian, which includes the coupling of neutrinos and axions. The zero order mass insertion, of the neutrino propagator, was corrected by adding one-loop terms depending on the neutrino and axion momenta. By combining the results of these corrections with current limits on the non-observation of the neutrinoless double beta decay, one can determine exclusion and allowed regions in the parametric space sustained by the strength of the axion-neutrino interaction, the axion mass and the couplings corresponding to left and right handed sectors of the electroweak currents. The results are compatible with neutrino masses smaller than a few tenths of eV.

## 5. Some Final Remarks

I have shared with Sabin Stoica many years of fruitful discussions in subjects of common interest, such as several approaches to the nuclear structure components of the

single and double beta decay models. In all cases, he has shown himself in a very direct and gentle mood. He did contribute largely to the success of a conference dedicated to double beta decay studies and also contributed through his continuous presence in journals. Thanks Sabin for all of these, and we hope you shall continue with these activities in the years to come.

**Funding:** This research received no external funding.

**Conflicts of Interest:** The author declares no conflict of interest

## Notes

1     Which is equivalent to the Mexican hat potential of the conventional Higgs mechanism.
2     See also these references for the meaning of the factors which appear in Equation (19), such as $\epsilon, \gamma, \beta, \zeta, \xi$ .

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
