# Peer review of "The Neutrino Mass Problem: From Double Beta Decay to Cosmology†"

_universe, doi:10.3390/universe9060275_

Round 1

Reviewer 1 Report

The work describes the effect of the axion mass and coupling on the neutrino mass. As such, this is an interesting topic and warrants publication. In particular, Fig. 1 is an interesting summary of the obtained results. In order to be more understandable to the readers, the manuscript needs a bit more work: 1) There is a typo on the right-hand side of Eq. (11); 2) In Eq. (18) the neutrino mass is not the physical one but effective one. This should be corrected and explained; 3) The curves of Fig. 1 are based on Eq. (17), is that right? If ye, this should be mentioned somewhere, maybe in the figure caption; 4) How do the different flavors come into play in Fig. 1? How are physical (bare) masses of different flavors determined? This cannot be straightforward since only squared mass differences are known from the neutrino-oscillation experiments; 5) Since the neutrino mass in Eq. (18) has to be the effective one, how do the masses of Eq. (17) [which is the main result of the present article, or is it?] connect to this effective effective mass <m_nu>? This should be carefully explained since this is the key to the connection of the axion physics to neutrinoless double beta decay.

Few comments on the language, mainly misprints: 1) Abstract, line 11: "extent" --> "extend"; 2) Page 2 4th line: The end of the sentence starting "The paper is devoted to the exploration..." is missing; 3) Line below Eq. (13): "li"? 4) Caption to Fig. 1 "mas" --> "mass"; 5) There is a wrongly spelled name in Ref. [4].

Reviewer 2 Report

This is a communication dedicated to Sabin Stoica 70th anniversary. I would recommend its publication after an improvement guided by the comments below:

- Review Eq.(15) and define the operation on the last term of the r.h.s.

- After Eq.(18), explain the values of ij in C_ij

- Some writing corrections:

Abstract: still has -> still have

Introduction: to the The conventional -> to the conventional

After Eq. 13: li and for spin-down one

General comment: A few sentences are a bit difficult to follow and can be improved. They could be divided in shorter sentences. For example:

- Abstract: "From the side of the nuclear structure and nuclear reactions, both theories and experiments, the need to elucidate these questions have, and still has, prompt crucial developments in the fields of double beta decay, double charge exchange and neutrino induced reactions."

- Introduction: "In at first glance different field, that is the search for the understanding of the composition of the dark matter in the Universe, did appear questions which are closely related to the double beta decay studies"

Correction needed:

- Pag. 2: "as well as to the The conventional"

- Pag. 4, after Eq. (13): "li and for spin-down one we have"

Reviewer 3 Report

The author proposes that the neutrinos may acquire their mass through interaction with a pseudoscalar axion field.  It’s an interesting paper and should be published.

Figure 1 shows the relationship between axion mass and the neutrino mass for two choices of the coupling constant g_a.  Three curves are identified in the legend as the masses of the electron, mu, and tau neutrinos.  Only two curves can be seen, but more important, those states do not have a mass.  Only the mass eigenstates nu_1, nu_2, and nu_3 have masses.  The figure should be replaced with a correct one, or some other explanation is needed.

Some minor points:

1.     P. 2 line 4, the The

2.     After Eq. 13 there is a spurious “li” beginning the line.

3.     Caption to Fig. 1 mas (this caption has to be revised for mass eigenstates anyway)

4.     Ref. 4 the second author is Petcov

Round 2

Reviewer 1 Report

The author has improved the manuscript to an extent that it is now publishable. Only one point: On p. 7 at two places is written "< m_v >". According to the adopted notation this should read "m_v". After this small correction, the manuscript is publishable.

Author Response

Dear Sir/Mrs
thanks for your comment about <m_nu>, it has been corrected.